# An Overview on the Performance of 1,2,3-Triazole Derivatives as Corrosion Inhibitors for Metal Surfaces

**DOI:** 10.3390/ijms23010016

**Published:** 2021-12-21

**Authors:** Meryem Hrimla, Lahoucine Bahsis, My Rachid Laamari, Miguel Julve, Salah-Eddine Stiriba

**Affiliations:** 1Laboratoire de Chimie Analytique et Moléculaire/LCAM, Faculté Polydisciplinaire de Safi, Université Cadi Ayyad, Sidi Bouzid, B.P. 4162, Safi 46000, Morocco; meryemhrimla.uca@gmail.com (M.H.); bahsis.lahoucine@gmail.com (L.B.); r.laamari@gmail.com (M.R.L.); 2Laboratoire de Chimie de Coordination et d’Analytique, Département de Chimie, Faculté des Sciences d’El Jadida, Université Chouaïb Doukkali, B.P:20, El Jadida 24000, Morocco; 3Instituto de Ciencia Molecular/*ICMol*, Universidad de Valencia, *C*/Catedrático José Beltrán 2, 46980 Valencia, Spain; miguel.julve@uv.es

**Keywords:** 1,2,3-triazole, click chemistry, corrosion inhibitor, metal, metal alloys, mechanism of inhibition

## Abstract

This review accounts for the most recent and significant research results from the literature on the design and synthesis of 1,2,3-triazole compounds and their usefulness as molecular well-defined corrosion inhibitors for steels, copper, iron, aluminum, and their alloys in several aggressive media. Of particular interest are the 1,4-disubstituted 1,2,3-triazole derivatives prepared in a regioselective manner under copper-catalyzed azide-alkyne cycloaddition (CuAAC) click reactions. They are easily and straightforwardly prepared compounds, non-toxic, environmentally friendly, and stable products to the hydrolysis under acidic conditions. Moreover, they have shown a good efficiency as corrosion inhibitors for metals and their alloys in different acidic media. The inhibition efficiencies (IEs) are evaluated from electrochemical impedance spectroscopy (EIS) parameters with different concentrations and environmental conditions. Mechanistic aspects of the 1,2,3-triazoles mediated corrosion inhibition in metals and metal alloy materials are also overviewed.

## 1. Introduction

Corrosion is a global problem, affecting growth in both developed and developing countries, that has adverse consequences which are not always visible, such as the lack of productivity due to industrial equipment and infrastructure deterioration, accidents, and significant economic losses. Industrial companies, including chemical, energy, transport, food processing, and construction are suffering from the devastation of corrosion-provoking losses of billions of dollars every year. The National Association of Corrosion Engineers (NACE) International reported that the global cost of corrosion is estimated at USD 2.5 trillion, which is equivalent to approximately 3.4% of the world’s Gross Domestic Product (GDP) [1,2,3,4]. Many industrial applications require the use of acidic solutions, such as industrial cleaning, removal of localized deposits, pickling, and other industrial preparative processes. As a matter of consequence of the aggressiveness of acidic media, corrosion inhibitors are frequently used to reduce the corrosive attack on metal-containing materials [5]. Corrosion inhibitors should be chosen based on the particular characteristics of the system, the type of the acidic medium, temperature of the solution, concentration, the presence of dissolved inorganic or organic substances, and in particular, the type of metallic materials. There are many types and compositions of corrosion inhibitors. For instance, organic compounds are adsorption-type inhibitors that adsorb onto the metal and prevent metal dissolution and reduction reactions [6,7]. Compared to inorganic corrosion inhibitors, the organic ones are less toxic, can be used at low concentrations, and have a better film-forming ability [8]. The adsorption properties of organic corrosion inhibitors are primarily linked to the presence of both π-electrons from aromatic rings and heteroatoms in the molecular structures [9]. Indeed, organic compounds containing nitrogen, phosphorus, sulfur, and oxygen atoms have been extensively investigated as corrosion inhibitors of metals and their alloys in acidic environments [9,10,11,12]. In this respect, triazole derivatives are the most used nitrogen-containing organic inhibitors, in particular the family of 1,2,3-triazoles prepared under the click chemistry concept. The interest in this class of heterocyclic compounds as structurally well-defined inhibitors of corrosion is justified by the increasing number of publications in the last two decades (Figure 1).

Click chemistry has received much attention in recent years, and it has become one of the most used concepts in the synthesis of new materials, surface modification, bioconjugation, polymer functionalization, and pharmaceutical development, among other fields [13,14,15,16,17,18,19]. It was first introduced by Sharpless in 2001 [20]. The click chemistry reactions have to be easy to carry out, afford high yields, produce only byproducts that can be removed by non-chromatographic methods, and stereospecific. Moreover, chemical reactions performed under such a regime are expected to be simpler by being insensitive to oxygen and water and simpler by product isolation [20,21]. Due to the simplicity and effectiveness of this concept, the click chemistry regime has spread widely to a variety of research fields, including drug discovery [18,22], biotechnology [23,24], materials chemistry [25], and nanomaterials [26,27], among many others [28,29,30]. The [3+2] cycloaddition reaction between azides and alkynes has been established as the prototype process among those classified as clickable reactions. This reaction that is thermodynamically favorable was first described by Huisgen in 1960 (Figure 2) and it is the most efficient preparative route for 1,2,3-triazoles [31,32]. This class of organic compounds can be widely used to obtain improved properties and so, new applications in research fields similar to corrosion protection, coatings, polymers and biomedical technology [5,33,34,35,36,37].

Envisaging corrosion applications, many 1,2,3-triazole derivatives have been reported as corrosion inhibitors and they have been found to possess corrosion-inhibiting effects for steels and their alloys, and for copper, iron, aluminum and their alloys in acidic solutions. It is also remarkable to note that the hybrids of these compounds have anticorrosive properties. In that respect, encapsulation of corrosion inhibitors is one of the promising approaches that can be used to separate inhibitors from the surrounding environment, and making them active and effective in the coating composition when needed [38]. In association with corrosion studies, the application of microcapsules is usually working as follows: (i) corrosion indicators that detect corrosion by changing visible [39,40] or fluorescence colors [41,42,43], and (ii) just as the corrosion inhibitors [44,45,46,47]. Koh et al. [48] prepared polyurethane (PU) microcapsules containing triazole derivatives by using interfacial polymerization of a diol-diisocyanate prepolymer and 1,4-butanediol (BD) as a chain extender aiming at self-healing coating against corrosion. An accelerated corrosion test indicated outstanding corrosion resistance with a rusting degree in the range of 0–0.008% via a self-healing mechanism through the use of the triazole derivative as an anti-corrosion substance. These findings reveal that triazole derivative-filled capsules have a great potential with potential industrial anti-corrosion applications.

Long-lasting anticorrosive coatings for steel have been developed by Joshi et al. [49] by using halloysite nanotubes loaded with three corrosion inhibitors: benzotriazole, mercaptobenzothiazole, and mercaptobenzimidazole. The corrosion protection efficiency was tested on ASTM A366 steel plates in a 0.5 M NaCl solution through the microscanning of corrosion current development and by studying paint adhesion. The best protection was found by using halloysite/mercaptobenzimidazole and benzotriazole inhibitors. Stopper formation with urea−formaldehyde copolymer was found to provide an additional increase in the corrosion efficiency as a result of the longer release of inhibitors.

## 2. Design and Synthesis of 1,2,3-Triazoles

The [3+2] cycloaddition reaction between organic azides and alkynes catalyzed by copper(I) (CuAAC) is the most applicable and best known synthetic method for the selective preparation of 1,4-disubstituted-1,2,3-triazole derivatives. The first described preparative route to obtain triazole compounds was reported in the 1960s by Huisgen et al. [50]. This process requires high temperatures, typically solvent (toluene or carbon tetrachloride) reflux conditions, and prolonged reaction periods generally between 12 and 60 h. Under these thermal conditions, the two possible regioisomers (1,4- and 1,5-disubstituted-1,2,3-triazole derivatives) are formed in an equimolar proportion (top route in Figure 2). All this changed in 2002 when Sharpless, Fokin, and their colleagues at the Scripps Research Institute (at La Jolla) and Meldal’s group at the Carlsberg Laboratory (at Copenhagen) [21,51,52] worked in parallel on this reaction and independently discovered that copper(I) catalyzed the azide-alkyne cycloaddition reaction (CuAAC), providing a regioselectivity synthesis of only 1,4-disubstituted-1,2,3-triazoles in very high yields (bottom route in Figure 2).

In contrast to the Huisgen process, the CuAAC reaction is simple to carry out, modular, very efficient, high yield, and it requires readily-available alkynes/azides as raw materials and reagents such as copper catalysts. Moreover, the CuAAC reaction is more than one hundred times faster at room temperature and it is easily achieved under these mild conditions. It requires removable or mild solvents such as water, thus meeting the requirements of “green chemistry”. All of these important characteristics are responsible for its large applicability in a variety of fields. The triazole derivatives obtained through this type of reaction (click chemistry) have been considered as alternative compounds to amides due to their stability under a wide range of conditions, including resistance to hydrolysis, oxidation, and reduction, conditions to which the amide bond is sensitive. These features increase their applications [53] and several 1,2,3-triazole derivatives also exhibit biological activity [54,55,56,57,58].

Depending on the nature of the azide intermediate (e.g., aliphatic or aromatic) and the starting material selected for the CuAAC reactions, different possibilities arise in this respect. Those found more frequently and which are summarized in Figure 1, involve the use of: (i) alkyl halides often of the “activated” type (e.g., benzyl, allyl, halocarbonyls) concerning the nucleophilic substitution reaction (Figure 1b); (ii) aromatic diazonium salts, either commercially available such as tetrafluoroborate salts or in situ produced by diazotization of the corresponding aromatic amine (Figure 1c); and (iii) epoxides that can lead to the formation of different substituted triazoles independently on the regioselectivity of the azide attack to the oxirane ring (Figure 1d).

In that respect, Alizadeh et al. described, for instance, an efficient click azide-alkyne [3+2] cycloaddition reaction for the synthesis of new 1-ester-4-sulfonamide-1,2,3-triazole derivatives (Figure 3). A synthesis that has been developed via a three-component reaction of *N*-propargylsulfonamides, sodium azide, and α-haloesters in a one-pot reaction (Figure 1b) [59]. They found that the method based on azidation of 2-bromoacetate with sodium azide, followed by copper-catalyzed azide-alkyne [3+2] cycloaddition reaction in H_2_O/EtOH as benign reaction medium, constitute the optimized experimental route to the desired triazole derivative (**3**) with a good yield.

Within the framework of green chemistry, Ali et al. used cetyltrimethylammonium bromide (CTAB) as an additive promoting the copper(I) iodide-catalyzed green and economical synthesis of 1,4-disubstituted-1,2,3-triazoles (Figure 1a). They initialized optimal CuAAC reaction conditions by using benzyl azide (**4**) with phenylacetylene (**5**) as a model substrate (see Figure 4). They carried out a preliminary experiment, where the azide-alkyne [3+2] cycloaddition reaction was performed in *t*-BuOH/H_2_O as a solvent with several sources of copper (1 mol%) as a catalyst, and CTAB as an additive at room temperature for 1 h. The reaction runs well under these conditions to yield the desired product (compound **6** in Figure 4) in a good yield, a feature that induced them to screen different commercial copper catalysts in the cycloaddition reaction with different solvents. In fact, copper(I) iodide was found to have high catalytic activity, resulting in a 75% conversion to 1,4-disubstituted-1,2,3-triazole, much faster in water than in organic solvents. When the process was performed in the presence of 10 mol% CuI using water as a solvent but in the lack of CTAB, the reaction did give low yield. Therefore, it is clear that CTAB is a very important additive in the performance of the CuAAC reaction that affords 1,4-disubstituted-1,2,3-triazoles with good to excellent yields [60].

Very recently, we have reported the synthesis of many 1,4-disubstituted-1,2,3-triazole compounds under the click chemistry regime by using copper(II) sulfate pentahydrate (1 mol%), sodium ascorbate (5 mol%), and water as a solvent at room temperature (Figure 5 and Figure 6) [61]. The 1,2,3-triazole products were obtained regioselectively as 1,4-isomer in excellent yields. (Figure 5 and Figure 6).

Several reaction conditions have been adapted for the synthesis of β-hydroxy-1,2,3-triazole 1,4-disubstituted (**9**, see Figure 7), such as high-temperature conditions, room temperature, or by the use of additives (Figure 1d) [62,63,64,65,66,67,68,69]. β-Hydroxy-1,2,3-triazole derivatives were synthesized by the [3+2] cycloaddition reaction between styrene oxide (**7**) and alkynes (**8**, see Figure 7), the 1,2,3-triazoles being obtained in good yields.

In this context, Souza et al. [70] and Chavan et al. [71] performed this reaction at room temperature. The first team carried out the reaction in the presence of copper(II) acetate, sodium ascorbate, and THF/H_2_O as a solvent mixture, while the second one used copper(I) iodide nanoparticles supported by bentonite clay BENTCuI nanoparticles, considered as an effective heterogeneous catalytic system in the high yield green synthesis of 1,2,3-triazoles by one-pot three-component reaction in water. Another strategy to prepare the desired 1,4-disubstituted 1,2,3-triazoles was adopted by Mishra et al. [69], which consisted of using an organo-photoredox catalyst Eosin Y, and EtOH:H_2_O as the reaction medium. The easy regioselective opening of the epoxides was followed by the [3+2] cycloaddition with alkynes under compact fluorescent light (CFL) irradiation as a source of visible light, resulting in the formation of the C–N bonds between azides and alkynes. Furthermore, Ghosh et al. [72] developed a new Cu(II)-Inorganic Co−Crystal (CuICC) as a versatile catalyst towards click chemistry for the preparation of 1,2,3-triazoles and β-hydroxy-1,2,3-triazoles, which acts as a heterogeneous catalyst for the synthesis of 1,2,3-triazoles. β-hydroxy-1,2,3-triazoles were obtained in a good yield by the CuICC-catalyzed reaction process without need for further column chromatographic purification.

## 3. 1,2,3-Triazoles as Corrosion Inhibitors

The continued research and development of new corrosion inhibitors is a matter of great importance and interest, both commercially and scientifically. Within this frame, several types of inhibitors have been developed and used to effectively inhibit the corrosion of metals; they can be classified as either inorganic or organic compounds. Typically, inorganic corrosion inhibitors have either anodic or cathodic action [73], while organic corrosion inhibitors have both anodic and cathodic roles (mixed-type), as well as a protective action by film adsorption (Figure 8). It is well established that the organic corrosion inhibitors are more effective and less expensive than the inorganic ones [74]. In this context, 1,2,3-triazole derivatives, which were initially studied as biologically active compounds and relevant medicinal compounds [55,56,75,76,77,78], have been reported as corrosion inhibitors for steels [79], copper, aluminum, and their alloys in corrosive media [80,81].

### 3.1. Steel and Its Alloys

Steel metal has many applications due to its good properties such as electrical and thermal properties and the possibility to protect this metal and its alloys against corrosion has attracted the attention of many research teams. The scope of this review focuses on published studies that have focused on the use of 1,4-disubstituted-1,2,3-triazoles as corrosion inhibitors. The steel type, corrosive medium, concentration of corrosion inhibitors, and the minimum and maximum reported values of inhibition efficiency IE (%) are listed in Table 1, including corrosion studies in different corrosive media for steel and its alloys (mild steel, carbon steel, and API 5L X52 steel) (Table 1). 

In this respect, Abdennabi et al. [82,83] reported the inhibitory effects of 1(benzyl)-4,5-dibenzoyl-1,2,3-triazole (BDBT) dealing with the corrosion activity of mild steel in acidic environments. They show that the corrosion rate of mild steel in 1% HCl was reduced by more than 95% in the presence of 50 ppm of BDBT. In addition, they noted that BDBT exhibits a mixed inhibition effect with a significant shift of the free corrosion potential towards the cathodic direction. Film persistence tests showed that BDBT forms a stable film on the electrode surface. Later, they reported on the structure/effect relationships that are effective in controlling the efficiency of the inhibition by variations in the inhibition characteristics that resulted from electron delocalization, steric effects, and lower solubility of the derivatives in the solvent’s exposure. Neupokoeva et al. [84] determined the inhibitory action of a series of 1,2,3-triazoles (Table 1) on mild steel in a solution of 0.05 M H_2_SO_4_. The galvanodynamic method was used to understand the electrochemical behavior of mild steel in a solution of H_2_SO_4_ in the presence of triazoles, and its additive compounds were studied in the temperature range 20–80 °C. They achieved the following results: (i) the studied 1,2,3-triazoles slow down the corrosion of mild steel in 0.05 M H_2_SO_4_, particularly at higher temperatures; (ii) the optimal inhibitor concentrations ensure a maximum protective effect; and (iii) all the investigated compounds are cathodic at a high temperature according to the mechanism of the inhibitory influence of the triazole compounds, which was examined. By using electrochemical methods, Babić-Samardžija et al. [85] studied the anticorrosion activity of 1-hydroxy benzotriazole, 1*H*-benzotriazole-1-methanol, and *N*-(1*H*-benzotriazol-1-ylmethyl)formamide in the corrosion of mild steel in 1 M HCl and 1 M HClO_4_ solutions (Table 1). Their results showed that these compounds are better inhibitors in 1 M HCl than in 1 M HClO_4_. However, Tafel’s behavior indicates that the inhibitors are predominantly mixed-type. According to the immersion time, the inhibiting efficiency increases with the increasing immersion time up to 16 h. The adsorption behavior of the investigated 1,2,3-triazoles was described by the Langmuir adsorption isotherm for the two acid media.

Sherif et al. [86] conducted electrochemical measurements to investigate the effect of 3-amino-5-mercapto-1,2,3-triazole (AMTA) (Table 1) on the corrosion process of steel in 2.0 M H_2_SO_4_ solution with exposure periods covering the range 3–180 min. According to the results of this study, the increase in exposure time leads to a remarkable strengthening of the steel corrosion by increasing the anode current, the cathode current, the corrosion current density and the corrosion rate. This effect also decreased both the solution and polarization resistances for the steel in the sulfuric acid solution. On the other hand, the increase in AMTA concentration from 1 × 10^−3^ to 5 × 10^−3^ M strongly reduced the steel corrosion by adsorption of the AMTA molecules on its surface, preventing its dissolution, as confirmed by surface analyses through EDX and SEM techniques.

The synthesis of new 1,4-disubstituted-1,2,3-triazole compounds that contain the phosphonic acid moiety was reported by Hrimla et al. followed by their anticorrosive activity on mild steel in 1 M HCl medium. The two triazole derivatives with phosphonic acid as pendant group, namely 3-(4-phenyl-[1,2,3]triazol-1-yl)-propyl-phosphonic acid diethyl ester (PTP) and 3-[4-(4-dimethylamino-phenyl)-[1,2,3]triazol-1-yl]-propyl]-phosphonic acid diethyl ester (DMPTP) [61], as well as 4-[1-(4-methoxy-phenyl)-1*H*-[1,2,3]triazol-4-ylmethyl]-morpholine (MPTM) [87], were evaluated by using weight loss and electrochemical techniques. The results showed that these compounds have an inhibitory efficiency greater than 90%, at a concentration of 5 × 10^−4^ M. They have also been found to act as mixed-type inhibitors and their adsorption on the surface of mild steel follows the Langmuir adsorption model. Besides, quantum chemical calculations studies predicted that the protonated forms of these triazoles could form an electrostatic interaction with the adsorbed chloride ions on the mild steel surface, whereas unprotonated molecules with lone pairs from the oxygen and nitrogen atoms, and π-electrons from the benzene ring, could fill the empty iron orbitals of the metal surface causing the chemisorption phenomenon.

In a systematic experimental study, Fernandes et al. observed that 1-benzyl-4-phenyl-1*H*-1,2,3-triazole (BPT) is an excellent corrosion inhibitor for mild steel in an HCl medium [104]. The polarization curves and thermodynamic parameters indicate that the triazole acts as a mixed inhibitor (chemisorption/physisorption) and Atomic Force Microscopy (AFM) shows the adsorption of a protective film of BPT molecules on the mild steel surface. The adsorption mechanism of the BPT molecules on mild steel can be explained by displacing water by the organic molecules Equation (1), electrostatic attraction, or formation of a metal-inhibitor complex, according to Equations (2) and (3).
BPT_(sol)_ + nH_2_O_(ads)_ → BPT_(ads)_ + nH_2_O_(sol)_
(1)
Fe_(s)_ → Fe^2+^_(aq)_ + 2e^−^(2)
Fe^2+^_(aq)_ + BPT_(ads)_ → [Fe-BPT]^2+^_(ads)_
(3)

Ma et al. reported that (1-benzyl-1*H*-1,2,3-triazole-4-yl)methanol (BTM) and PTM exhibit higher inhibition efficiency in HCl for mild steel surfaces [9]. Thermodynamic parameters confirm from one side that triazole derivatives act as mixed inhibitors and from another side, AFM and SEM analysis show the formation of a protective film, confirming the adsorption of BTM and PTM on the metal surface. Computational chemistry studies using DFT calculations and MD point to the fact that the adsorption process of triazole derivatives on the mild steel surface can occur by sharing the lone pair electrons of the nitrogen atoms with the vacant orbitals of the iron atoms, or by accepting electrons from the iron surfaces. Moreover, the interaction of the pyridine fragment of PTM with the iron surface is stronger than that of the benzene one of BTM. This led to the parallel interaction of PTM molecules to the mild steel surface, which conducts to a better-blocking effect of PTM for the mild steel surface compared to the BTM.

The addition of the theophylline group to the 1,2,3-triazole moieties demonstrated that the inhibition activity of the theophylline-triazole derivatives is greater than the inhibition activities of theophylline and theobromine, as reported by Espinoza-Vázquez et al. [100]. The adsorption study for *N*-(1*H*-benzotriazol-1-ylmethyl)formamide showed that the corrosion inhibition of the API 5 LX52 steel process follows the Langmuir isotherm with a combined physisorption-chemisorption process. The atomic charges reveal that both oxygen atoms of the theophylline have a high negative charge indicating that these sites are electron-donating atoms in the interaction with the metallic surface.

The corrosion efficiency of BTA on the carbon steel surface was investigated and its mechanism of corrosion inhibition was studied by XPS and FTIR [107]. A strong bond was formed between the azole (N=N) of the triazole moieties and the steel surface, leading to a 2 nm thickness film of iron-azole complexes, which can inhibit the steel corrosion in neutral aqueous electrolytes containing chloride ions. Moreover, the iron-azole layer located at the interface acts as an adhesion promoter, increasing the interaction of polymeric coatings with the steel surface.

### 3.2. Copper and Its Alloys

Copper and its alloys have attracted considerable interest due to their excellent thermal and electrical conductivity and they have been widely applied for the storage of drinking water in buildings and houses, as well as for many different fluids ranging from oil and chemical processes to marine industries [108,109]. However, it is an active metal that does not resist corrosion enough, especially in the presence of chloride anions [110]. Therefore, several of the solutions were tested to protect copper and its alloys from corrosion [10,111,112]. In this regard, triazole derivatives such as benzotriazole (BTA) are the first 1*H*-1,2,3-triazole compounds investigated as corrosion inhibitors for copper and its alloys and so they are classified as effective corrosion inhibitors for copper surfaces [113,114,115,116]. Villamil et al. [117] found that the SDS-BTAH interaction at the Cu/H_2_SO_4_ interface in aqueous solutions can provide good inhibition of copper corrosion. Bi et al. [118] also studied the inhibition of copper in tap water in the presence of BTA (Table 2) by a cyclic voltammetric curve, concluding that the inhibitor forms a polymerized reactive surface film. The mechanism of BTA inhibition has generally been attributed to the adsorption of single BTA molecules onto the copper surface and the formation of a Cu(I)-BTA polymer film. The inhibition of copper corrosion has been studied for decades and it has most effectively been achieved with BTA, which does not show any particular toxicity [109]. Cheng Shi et al. [119] formed triazole films on the copper surface from a chemical click reaction between 2,2-dimethylethynyl carbinol (MBY) and tosyl azide (TA), to study the effect of the copper(I) ions on the assembled films by using electrochemical measurements. They found that the assembled films had the highest protection efficiency, ca. 93.8%. Chloride ion etching produces copper(I) ions, which catalyze the formation of compact triazole films. They also found that the assembled films exhibit anodic passivation and prevent anodic corrosion reactions.

Later, Yu et al. [34] performed in situ assemblies of the 2-(1-tosyl-1*H*- 1,2,3-triazole-4-yl) propane-2-ol (TTP in Table 2) monolayer on the copper surface by click reaction between MBY and TA. Electrochemical results indicate that the clicked TTP film can strongly reduce the corrosion rate of copper in 3wt% NaCl and it has a protection efficiency of 93.6%. On the other hand, the polarization curves show that self-assembled TTP is an anodic passivation type inhibitor. Li et al. [10] synthesized 1,4-disubstituted-1,2,3-triazole inhibitors via click reaction of TTE (Table 2) and TTP to manufacture a self-assembled membrane (SAM) of synthetic TTP and TTE on the copper surface. As a result, they found that the SAM of TTE and TTP offers good protection to copper in 3wt% NaCl and it mainly delays the anodic reaction, with the SAM of TTP having a better inhibition performance than the one of SAM of TTE (93.1% and 89.4%, respectively). This is likely due to the existence of two methyl groups in the TTP molecule. Both triazoles exhibit effective inhibition properties for copper in 3 wt% NaCl solution. Their adsorption processes were explained by the electron donation-withdrawing interaction between the copper atom and the inhibitor molecules. Furthermore, quantum chemistry calculations and MD simulations revealed that the oxygen atoms and the nitrogen atoms of the triazole moiety are primarily responsible for the adsorption of these two inhibitors on the copper metal surface. Zhang et al. [120] have developed an environmentally benign and practical method combining self-assembly technology and click chemistry reaction, which is considered an innovation in the click assembly process to protect copper materials. However, the XPS analysis showed that the addition of sodium thioacetate (STG) could reduce the values of the copper(II) ratio on the metal surface, which conducts to the lower surface oxidation, according to Equation (4) [121], where RSSR represents the reaction product containing the -S-S- bond.

These findings confirm that STG can form a complex with copper(I) and then lead to increase the contents of this univalent cation on the copper surface This feature promotes the click CuAAC reaction and the formation of a protective film of TTE on the copper surface using the nitrogen atom from the triazole ring and the oxygen atoms from the TTE molecules.
2STG + 2Cu(II) → RSSR + 2Cu(I) + 2H^+^
(4)

Wang et al. investigated the inhibition mechanism of a variety of new dibenzotriazoles bearing alkylene linkers on the copper surface using the Langmuir isotherm, FT-IR spectroscopy, and XPS [125]. The results obtained suggest that the adsorption equilibrium concentration of the inhibitors would play an important role on the corrosion inhibition. The XPS analysis reveals that the target inhibitors are adsorbed on the copper surface mainly through the formation of chemical bonds. The formation of a copper(I)-benzotriazole complex by using nitrogen atoms from two benzotriazole moieties can be deduced by the XPS results. Moreover, the quantum chemical parameters of the highest occupied and lowest unoccupied molecular orbitals (HOMO and LUMO, respectively) of the dibenzotriazole derivatives and their energy gap (ΔE) can be used to explain the higher inhibition efficiencies of these organic inhibitors.

### 3.3. Aluminum and Its Alloys

Aluminum and its alloys have important industrial and economic values because of their good thermal and electrical conductivity and low cost and weight, which is only one-third the density of steel [126,127]. The most important property of aluminum is its ability to effectively resist atmospheric and aqueous solutions, due to the fast formation of a protective thin film of aluminum(III) oxide, which prevents the corrosion of the metal in a corrosive environment. However, aluminum undergoes localized corrosion in contact with corrosive solutions, making the use of corrosion inhibitors one of the most effective methods of protection against corrosion of metals and alloys. To address this phenomenon, many organic compounds have been used as corrosion inhibitors of aluminum and its alloys [80,128,129,130,131], except the organic 1,4-disubstituted 1,2,3-triazoles, which have unfortunately been the subject of very few publications as corrosion inhibitors of this type of metal. In this respect, an investigation on the use of tolyltriazole (TTA) as a corrosion inhibitor against corrosion of Al/Cu, Al/Si/Cu, and Al/Cu/Fe alloys in HCl (pH = 0.5) and NaCl (pH = 6 and 11), respectively, was carried out by Önal et al. at 15, 25, and 35 °C (Table 2) [12]. The results have shown that the inhibition efficiency of TTA changed with the pH and temperature. Furthermore, TTA was more efficient at pH = 0.5 and 6 than at pH = 11. Increasing the temperature from 15 °C to 35 °C decreased the inhibition effects of TTA. However, TTA has been adsorbed on the alloys to form a Cu(I)-TTA film, and it also acts as a cathodic inhibitor. A report by Zheludkevich et al. in 2005 [132] dealt with the evaluation of triazole and thiazole derivatives as corrosion inhibitors of the 2024 aluminum alloy. In fact, the inhibition performance of benzotriazole in comparison with other thiazole derivatives on the corrosion of the 2024 aluminum alloy was studied in neutral chloride solutions (Table 2). The corrosion protection performance was investigated by means of DC polarization and electrochemical impedance spectroscopy (EIS). The results revealed that benzotriazole offers better corrosion protection in comparison to the other tested compounds, decreasing the rate of the anodic reaction and increasing the Tafel slope by approximately three times. However, it prevents the process of dealloying of the intermetallic particles and copper dissolution causing deceleration of the oxygen reduction.

Nazeer et al. [123] explored the inhibition effectiveness of 5-methyl-1*H*-benzotriazole (MBT) on the corrosion of α-aluminum bronze in clean and sulfide-polluted salt water, using the potentiodynamic polarization, electrochemical impedance spectroscopy (EIS), and electrochemical frequency modulation (EFM) (Table 2). These different techniques have confirmed that MBT acts as a good corrosion inhibitor of the aluminum bronze corrosion in unpolluted and sulfide-contaminated salt water. The presence of sulfide ions (2 ppm) of MBT decreases the inhibition efficiency of MTB against α-aluminum bronze in chloride solutions from 94.7% to 89% at an MTB concentration of 5 × 10^−4^ M. The PDP measurements showed that MBT behaves as a mixed-type inhibitor.

Recently, Balaskas et al. [37] and Marcelin et al. [124] investigated the inhibition of the corrosion of AA 2024-T3 2024 and AA2024 aluminum alloys in 3.5% NaCl solution and neutral aqueous solution, respectively, using 8-hydroxyquinoline (8-HQ) and BTA (Table 2). In parallel, the inhibition efficiency of 2-mercaptobenzothiazole (2-MBT) was further investigated by Balaskas et al. and compared to that of 8-HQ and BTA (Table 2). In fact, it was found that 2-MBT has higher inhibition efficiency compared to the other two inhibitors, maintaining high values of the noise resistance for over 400 h of immersion and high low-frequency impedance values in 3.5wt% NaCl solution. Moreover, electrochemical techniques suggest that the 2-MBT inhibitor decreases the rate of both the anodic and cathodic reactions. Importantly, Marcelin et al. showed that the 8-HQ and BTA mixture had a synergistic effect on the effective corrosion protection of AA2024. The electrochemical result confirmed that the galvanic coupling responsible for the corrosion process is strongly limited in the presence of both compounds, which is not the case when the inhibitors are used separately. This electrochemical result was confirmed by SEM analysis.

New poly 1,4-disubstituted-1,2,3-triazole derivatives were reported by Armelin et al. by using a variety of alkyne and azide monomers and the resulting polymers were used as precursors to anti-corrosion coatings for a standard high-strength AA2024 aluminum alloy [130]. FTIR, DSC, pull-off testing, and EIS were used to prove that these materials exhibit a remarkable resistance to corrosive pitting due to the combination of three complementary properties: highly cross-linked films, good adhesion, and excellent impermeability. Furthermore, the presence of several methylene groups on the azide compositions helps to moderate the degree of cross-linking among the polymeric chains, offering beneficial flexibility to the polymer film on the surface of the aluminum alloy.

### 3.4. Effect of the Structure of the Inhibitors on Their Inhibition Efficiency

As electron-rich aromatic heterocyclic systems, triazoles can bind to mild steel through both nitrogen atoms and aromatic triazole ring, including through additional aromatic substituents when present. The adsorption strength does largely depend on (i) the nature of the functional group present in the triazole derivative (ii) the structure of the rest of the molecule, as well as (iii) on the electron-donating/withdrawing elements, which indicates that it is not the only triazole moiety that interacts with the metal surface. Therefore, a real strategy to increase the number of adsorption sites consists of increasing the number and nature of the lateral substitutions, particularly the functional groups, on the triazole moiety, which has been found to influence its ability to prevent corrosion phenomena in a strong way; a fact that can be observed on the recorded inhibition efficiency of heterocyclic molecules (see Table 1 and Table 2). For instance, Espinoza Vázquez et al. [133] reported the synthesis and evaluation of 1,2,3-triazole derived from para-substituted benzyl alcohol as acid corrosion inhibitors for API 5L X52 steel through several experiments. The results showed that all the derivatives with different substituents on the benzyl of the triazole unit presented better inhibition capacity than 1,2,3-triazole derived from unsubstituted benzyl alcohol. In fact, electrochemical impedance spectroscopy combined with polarization curves revealed that 1,2,3-triazole derivative that contains bromine substituent in its structure has the best inhibition efficiency when compared to benzyl alcohol and its other derivatives, reaching a maximum of 94.2% at 50 ppm. 1,2,3-triazole compound which has chlorine in its structure showed the lowest inhibition efficiency, reaching its maximum at 100 ppm with 88.1%. These results were attributed to the structure of the inhibitive molecule, i.e., the presence of a pair of free electrons of the halogen can also interact with the metal surface, a fact that has also been supported by a theoretical study. The DFT study revealed that 1,2,3-triazole derived from para-bromobenzyl alcohol exhibited the greatest charge transfer to the iron surface. The calculated free binding energies were close to those experimentally determined by the standard adsorption free energies. Besides, all corrosion inhibitors provided a protective layer against electrophilic and nucleophilic attacks, as confirmed by Fukui functions.

## 4. Mechanism of Corrosion Inhibition

The mechanism of corrosion inhibition involves two steps. The first step consists of the transfer of the corrosion inhibitors over the metallic surface, whereas the second step includes the interactions between the adsorbed inhibitor molecules and the metal surface, which can be electrostatic or chemical type interactions [134,135,136,137]. These steps are primarily driven by the electronic structure of the inhibitor molecules and the charges on the metallic surface. The electrostatic interaction (physisorption) mainly results from the interaction between charges of opposite sign over the metal surface and the inhibitor molecule, whereas the chemisorption is due to the direct contact of the metal surface with the inhibitor molecules, sharing electrons between them. The net charge on the surface of the metal is zero at the potential of zero charges (PZC) [138]. For a negative charge surface with respect to PZC, the adsorption of the cationic moiety occurs (protonated inhibitor molecules), while the adsorption of the anionic moiety (Cl^−^ and SO_4_^2−^) takes place at the positive charge surface with respect to PZC.

Triazoles are among the most efficient corrosion inhibitors and their adsorption onto the surface of the metal involves the heteroatoms and the π-electrons of the aromatic system, which have an important role in the adsorption process. In fact, the triazole derivatives interact with the metal surface through donor-acceptor interactions between the –N=N–, –C=N–, and (–C=C–) π-electrons of the benzene rings and the empty-orbitals of the metal atoms. The adsorption mechanisms of triazole molecules at the metal surfaces are classified into four types: chemisorption, electrostatic adsorption (physisorption), retrodonation (involving the d-orbitals of the surface metal atoms and the unoccupied molecular orbitals of the inhibitor molecules), and the formation of the complex metal-inhibitor [105,137,139,140] (Figure 2).

## 5. Conclusions

This review article has overviewed the recent advances on the corrosion inhibition properties of clickable 1,2,3-triazoles for steels, copper, aluminum, and theirs alloys in several corrosive media. A variety of 1,2,3-triazole inhibitors with different electronic and steric modulations were discussed taking into account their IE% values in different corrosive media and concentration of the inhibitors. In most of the reports, the corrosive environment concentration tested was 1 M HCl, with mild steel as the largely utilized metal surface in many industrial fields. The inhibition mechanism action of 1,2,3-triazoles has generally been attributed to the formation of a surface layer or film formed after adsorption of the 1,2,3-triazole species on the metal surface, involving reactions at the interfaces between the corrosive electrolyte and the metal, although other studies have indicated that this film is made of inhibitor-containing metal complexes. Further studies focusing on mechanistic aspects of the action of these structurally well-defined heterocycles are necessary in order to improve their rational design for real industrial trials that involve corroded metal surfaces. In addition, 1,2,3-triazoles-based polymers deserve to be studied for their potential corrosion inhibition since they offer a large surface of action on metal surfaces.

## Data Availability

Not applicable.

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
