# Peer review of "An Overview on the Performance of 1,2,3-Triazole Derivatives as Corrosion Inhibitors for Metal Surfaces"

_ijms, 2021, doi:10.3390/ijms23010016_

Round 1

Reviewer 1 Report

Manuscript Number: ijms-1494388:

An Overview on the Performance of 1,2,3-Triazole Derivatives as Corrosion Inhibitors for Metal Surfaces

General comment

The paper presents a review account for the most recent and significant research results from the literature on the design and synthesis of 1,2,3-triazole compounds and their corrosion inhibition performance for steels, copper, iron, aluminum, and their alloys in several aggressive media. These triazole derivative compounds have attracted wide interest from many researchers because of their exceptional properties. The review contains necessary and interesting data in the field.

Some recommendations and observation are listed below:

  1. The important aspects are reached but they are not sufficiently explicit or systematized.

  1. As aromatic systems, rich in electrons, triazole derivatives are able to bind to MS through weak interactions such as cation-π, π-π stacking, ion-dipole, coordination bonds, hydrogen bonds, van der Waals forces or hydrophobic effect. It is beneficial to explain in more detail, the influence of the structure of the inhibitor that brings new or makes them better than another. The bonding strength is largely dependent on the nature of the functional group in and also on the structure of the rest of the molecule. For example, it was reported that the nature of side substitutions of the triazole moiety has strongly influenced its ability to prevent corrosion phenomena. The recorded inhibition efficiency of some heterocyclic molecules depends on the substituent nature [ex International Journal of Electrochemistry. 2019;2019:6759478]. The introduction of other functional groups and conjugated systems is a strategy aims to introduce more adsorption sites. Is necessary to discuss these aspects for the selected inhibitors.

  1. The literature also states that a number of hybrids of these compounds also have anticorrosive properties. It would be good to include them in the introduction. Encapsulation of corrosion inhibitors is beneficial and the inhibitor can diffuse slowly outside the host material to ensure continuous delivery of inhibitors to the corrosion sites and thus provide long-term protection against corrosion.

  1. What is the reason for paragraphs 3.4., 3.5 and 3.6 Sections 3.1, 3.2 and 3.3 talk about steel, copper and aluminum and their alloys and the inhibitory effect of compounds? It would be more desirable to rename the paragraphs (Experimental and DFT Investigation on Corrosion ...) or to include their content in the related paragraphs.

  1. Detailed for table 1 the complete caption, that include the substrate also. Insert under the table the notation of used substrate abbreviation (MS, CS….)

  1. The temperature is important) and the values are missing in tables. (IE and CR depends on it).

  1. Is better for a clearer image, that the concentrations presented in the tables should be recalculate and presented in the same mode (ppm or molar).

  1. R 254. "The results showed that these compounds have an inhibitory efficiency of more than 90%" Specify the concentration because at a lower concentration IE is below 90%.

  1. Rewrite paragraph 3.5 for a better understanding, it's a little confusing.

  1. Some space, comma must be corrected. References must be revised and write in the same manner.

Author Response

First of all, we would like to thank both reviewers for appreciating our research work and for their positive reports on our manuscript ref. ijms-1494388.

Please find hereunder our “point-by-point” responses to the comments made by the reviewers. The new additions and corrections are noted in red color in the main text of the manuscript.

Point-by-point responses to the reviewers:

Reviewer 1:

1. The important aspects are reached but they are not sufficiently explicit or systematized.

Authors: The important aspects on the use of 1,2,3-triazoles as corrosion inhibitors for metallic surfaces that has been addressed in this article review are now explicitly explained through the responses to the different points that both reviewers commented.

2. As aromatic systems, rich in electrons, triazole derivatives are able to bind to MS through weak interactions such as cation-π, π-π stacking, ion-dipole, coordination bonds, hydrogen bonds, van der Waals forces or hydrophobic effect. It is beneficial to explain in more detail, the influence of the structure of the inhibitor that brings new or makes them better than another. The bonding strength is largely dependent on the nature of the functional group in and also on the structure of the rest of the molecule. For example, it was reported that the nature of side substitutions of the triazole moiety has strongly influenced its ability to prevent corrosion phenomena. The recorded inhibition efficiency of some heterocyclic molecules depends on the substituent nature [ex International Journal of Electrochemistry. 2019;2019:6759478]. The introduction of other functional groups and conjugated systems is a strategy aims to introduce more adsorption sites. Is necessary to discuss these aspects for the selected inhibitors.

Authors:  A new subsection (3.4.) that addresses the functional groups effect on the the adsorption and corrosion inhibition properties of 1,2,3-triazoles is added (lines: 459-487).

3. The literature also states that a number of hybrids of these compounds also have anticorrosive properties. It would be good to include them in the introduction. Encapsulation of corrosion inhibitors is beneficial and the inhibitor can diffuse slowly outside the host material to ensure continuous delivery of inhibitors to the corrosion sites and thus provide long-term protection against corrosion.

Authors: Following the reviewer’s suggestion, additional statements from the literature on the properties of hybrids material as anticorrosive inhibitors were added in the introduction section (lines 82-103).

4. What is the reason for paragraphs 3.4., 3.5 and 3.6 Sections 3.1, 3.2 and 3.3 talk about steel, copper and aluminum and their alloys and the inhibitory effect of compounds? It would be more desirable to rename the paragraphs (Experimental and DFT Investigation on Corrosion ...) or to include their content in the related paragraphs.

Authors: The main text has now been reorganized and appears as follows: 1. Introduction; 2. Design and Synthesis of 1,2,3-triazoles; 3. 1,2,3-Triazoles as Corrosion Inhibitors (with three sections related to three different metal surfaces and their alloys and a fourth section ( 3.1.4) that corresponds to the Effect of the structure of the inhibitors on the inhibition efficiency, ending with  a section entitled 4. Mechanism of Corrosion Inhibition.

5. Detailed for table 1 the complete caption that includes the substrate also. Insert under the table the notation of used substrate abbreviation (MS, CS….)

Authors: Detailed data for Table 1 have been appropriately quoted under Table 1 (R 328).

6. The temperature is important) and the values are missing in tables. (IE and CR depends on it).

Authors: The working temperature is now specified (see Tables 1 and 2).

7. Is better for a clearer image, that the concentrations presented in the tables should be recalculate and presented in the same mode (ppm or molar).

Authors: The concentrations presented in the tables have been included in the same mode (ppm).

8. R 254. "The results showed that these compounds have an inhibitory efficiency of more than 90%" Specify the concentration because at a lower concentration IE is below 90%.

Authors: The concentration is now specified in this revised version (line 284).

9. Rewrite paragraph 5 for a better understanding, it's a little confusing.

Authors: The rephrasing of paragraph 2.5, previously 3.5 has been done.

10. Some space, comma must be corrected. References must be revised and write in the same manner.

Authors: Spelling, text edition and references style, all these points have been thoroughly checked in the present revised version.

Reviewer 2 Report

Review on the article “

 An Overview on the Performance of 1,2,3-Triazole Derivatives  as Corrosion Inhibitors for Metal Surfaces An Overview on the Performance of 1,2,3-Triazole Derivatives 2 as Corrosion Inhibitors for Metal Surfaces” submitted for publication to MDPI  Int. J. Mol. Sc.i.

The article reviews the last investigations devoted to the synthesis of different derivatives of 1,2,3 triazole and corrosion inhibiting properties. The interaction of different technical metals and alloys with azoles was discussed. The mechanisms of corrosion inhibition were pointed out.

My main conclusion as the reviewer is that the article is written well, on a good level and can be published.

For me may be more interesting is comparison of different azoles and impact of functional groups like toile and the chelate effect of complex formation on the inhibiting power.

However the article as it is also useful.

The small comments                                                        

Line 455

“contact of the metal surface with the inhibitor molecules, sharing electrons between them. The net charge on the surface of the metal is zero at the potential of zero charges (PZC) [137]. F contact of the metal surface with the inhibitor molecules, sharing electrons between them. The net charge on the surface of the metal is zero at the potential of zero charges (PZC) [137].”

In conclusions, you wrote that formation of the layer containing inhibitor and dissolving metal cations is the main reason of inhibition. It is right. With time, the power of inhibitor increases due to growth of the film. Thus in mechanism is mainly involved formation of the non-soluble salt like layer. Otherwise, you are contradicting to conclusion.

482 “mechanism action of 1,2,3-triazoles has generally been attributed to the formation of a surface layer or film formed after adsorption of 1,2,3-triazole species on the metal surface,”

478  “IE% values, which were evaluated using Electrochemical Impedance Spectroscopy parameters”

You did not measure EIS, please rephrase the sentence.

Author Response

Reviewer 2.

The article reviews the last investigations devoted to the synthesis of different derivatives of 1,2,3 triazole and corrosion inhibiting properties. The interaction of different technical metals and alloys with azoles was discussed. The mechanisms of corrosion inhibition were pointed out.

My main conclusion as the reviewer is that the article is written well, on a good level and can be published.

For me may be more interesting is comparison of different azoles and impact of functional groups like toile and the chelate effect of complex formation on the inhibiting power.

However the article as it is also useful.

Authors:  We thank this reviewer for appreciating this article review and for recommendation of its publication. We also agree with him/her that an account comparing the anticorrosive properties of the heterocycle compounds that belong to the azoles could be more interesting.

The small comments                                                        

Line 455 “contact of the metal surface with the inhibitor molecules, sharing electrons between them. The net charge on the surface of the metal is zero at the potential of zero charges (PZC) [137]. F contact of the metal surface with the inhibitor molecules, sharing electrons between them. The net charge on the surface of the metal is zero at the potential of zero charges (PZC) [137].”

In conclusions, you wrote that formation of the layer containing inhibitor and dissolving metal cations is the main reason of inhibition. It is right. With time, the power of inhibitor increases due to growth of the film. Thus in mechanism is mainly involved formation of the non-soluble salt like layer. Otherwise, you are contradicting to conclusion.

482 “mechanism action of 1,2,3-triazoles has generally been attributed to the formation of a surface layer or film formed after adsorption of 1,2,3-triazole species on the metal surface,”

Authors: This statement has been now corrected (see lines 519-523).

478  “IE% values, which were evaluated using Electrochemical Impedance Spectroscopy parameters”

You did not measure EIS, please rephrase the sentence.

Authors: This comment has been taken into consideration and the corresponding sentence rephrased (line 540).

Round 2

Reviewer 1 Report

Manuscript Number: ijms-1494388:

An Overview on the Performance of 1,2,3-Triazole Derivatives as Corrosion Inhibitors for Metal Surfaces.

General comment

The paper presents a review account for the most recent and significant research results from the literature on the design and synthesis of 1,2,3-triazole compounds and their corrosion inhibition performance for steels, copper, iron, aluminum, and their alloys in several aggressive media. The review contains necessary and interesting data in the field.

The authors responded to each comment and made the requested changes.

Some recommendations and observation are listed below:

  1. At R 228 the authors write "values of EI are listed in Table 1". What is EI?
  2. In table 1 the column attributed to the abbreviation is filled with the entry number. Insert the abbreviation for selected compounds. Ex: In table 1 at entry 10 Abb./Symbols should be (MPTM). Check all these abbreviations and fill in the tables with the appropriate one for each compound

Remove the square comma after [88][.

3. At R 354 author mentioned a compound from table 3. This table is missing, insert or correct the table number.

The reference [40] and [142] are not presented in table 2, as author mentioned at R 427. Also in Table 2 the Abb./Symbols must be corrected as was request above, for Table 1.

Ex: In table 3 entry 5 Abb./Symbols should be MBT

4. At R 422 the author write “…exhibited 94.9 and 89.2% % of…”

Delete one %. To correlate the text with the value presented in Table 2, it is recommended to specify the level of concentration.

 Ex: exhibited 94.9 and 89.2% of inhibition at concentration higher than….  ppm.

5. At R456 correct …charge transfert

 At R484 correct  the reference number [146-1149].

6. Check the all references. Eliminate unnecessary space, add comma after volume number, add space where is missing.

Some examples No 5:  J. Mater. Environ. Sci. 2018, 9 (2), 453-465.

 No 54 A dv. Synth. Catal. 2017,359(2), 202–224

No 73, Catal. Lett. 2017,147(10), 2600–2611

No.77 Corros. Inhibitors. Principles. Recent. Applications.2018, 1, 79-94

No.109  RSC. Adv. 2016, 6(77):72885–72896.

R123 RSC. Adv. 2016, 6(77):72885–72896.

Author Response

Manuscript Number: ijms-1494388:

An Overview on the Performance of 1,2,3-Triazole Derivatives as Corrosion Inhibitors for Metal Surfaces.

General comment. The paper presents a review account for the most recent and significant research results from the literature on the design and synthesis of 1,2,3-triazole compounds and their corrosion inhibition performance for steels, copper, iron, aluminum, and their alloys in several aggressive media. The review contains necessary and interesting data in the field.

The authors responded to each comment and made the requested changes.

Authors: Once again, we thank this reviewer for the comments and observations that really helped improving this review article. The new addition/corrections made in light of the reviewer comments are in red color and yellow shadow.

Some recommendations and observation are listed below:

1. At R 228 the authors write "values of EI are listed in Table 1". What is EI?

Authors: EI reads now IE (%), which is the abbreviation of Inhibition Efficiency (line 232 and Tables 1 and 2).

2. In table 1 the column attributed to the abbreviation is filled with the entry number. Insert the abbreviation for selected compounds. Ex: In table 1 at entry 10 Abb./Symbols should be (MPTM). Check all these abbreviations and fill in the tables with the appropriate one for each compound

 Authors Abbreviations of triazole derivatives shown in Table 1 and 2 are given, when and as reported in the corresponding reference.

Remove the square comma after [88][.

Authors: Done.

3. At R 354 author mentioned a compound from table 3. This table is missing, insert or correct the table number.

The reference [40] and [142] are not presented in table 2, as author mentioned at R 427. Also in Table 2 the Abb./Symbols must be corrected as was request above, for Table 1.

Ex: In table 3 entry 5 Abb./Symbols should be MBT

Authors: Table 3 reads now Table 2. Abbreviations of triazole derivatives shown in Table 1 and 2 are given, when and as reported in the corresponding reference.

4. At R 422 the author write “…exhibited 94.9 and 89.2% % of…”

Delete one %. To correlate the text with the value presented in Table 2, it is recommended to specify the level of concentration.

 Ex: exhibited 94.9 and 89.2% of inhibition at concentration higher than….  ppm.

Authors: The corresponding paragraph that corresponds to corrosion inhibition efficiency results was explicitly rephrased (lines 429-433).

5. At R456 correct …charge transfert

Authors: Done.

At R484 correct  the reference number [146-1149].

Authors: Done.

6. Check the all references. Eliminate unnecessary space, add comma after volume number, add space where is missing.

Some examples No 5:  J. Mater. Environ. Sci. 2018, 9 (2), 453-465.

 No 54 A dv. Synth. Catal. 2017,359(2), 202–224

No 73, Catal. Lett. 2017,147(10), 2600–2611

No.77 Corros. Inhibitors. Principles. Recent. Applications.2018, 1, 79-94

No.109  RSC. Adv. 2016, 6(77):72885–72896.

Authors: Done.